# StructGPT: A General Framework for Large Language Model to Reason over Structured Data

**Jinhao Jiang[1,3][*], Kun Zhou[2,3][*], Zican Dong[1], Keming Ye[4],**
**Wayne Xin Zhao[1,3][†] and Ji-Rong Wen[1,2,3]**
[1]Gaoling School of Artificial Intelligence, Renmin University of China.
[2]School of Information, Renmin University of China.
[3]Beijing Key Laboratory of Big Data Management and Analysis Methods.
[4]University of Electronic Science and Technology of China.
jiangjinhao@ruc.edu.cn, batmanfly@gmail.com

## Abstract

In this paper, we aim to improve the reasoning ability of large language models (LLMs) over structured data in a unified way. Inspired by the studies on tool augmentation for LLMs, we develop an *Iterative Reading-then-Reasoning (IRR)* framework to solve question answering tasks based on structured data, called **StructGPT**. In this framework, we construct the specialized interfaces to collect relevant evidence from structured data (*i.e., reading*), and let LLMs concentrate on the reasoning task based on the collected information (*i.e., reasoning*). Specially, we propose an *invoking-linearization-generation* procedure to support LLMs in reasoning on the structured data with the help of the interfaces. By iterating this procedure with provided interfaces, our approach can gradually approach the target answers to a given query. Experiments conducted on three types of structured data show that StructGPT greatly improves the performance of LLMs, under the few-shot and zero-shot settings. Our codes and data are publicly available at https://github.com/RUCAIBox/StructGPT.

## 1 Introduction

Recently, large language models (LLMs) (Brown et al., 2020; Zhao et al., 2023) have made remarkable advancements in the NLP field. Existing work (Ouyang et al., 2022a; Zhang et al., 2022) has demonstrated that LLMs (*e.g.,* ChatGPT or GPT-4 (OpenAI, 2023)) have strong zero-shot capability to solve a broad range of tasks using specially designed prompts, without task-specific fine-tuning.

Despite the successes, recent work has also revealed that LLMs may generate unfaithful information in conflict with the factual knowledge (Li et al., 2023), and also fall short of mastering domain-specific or real-time knowledge (Schick et al., 2023; Peng et al., 2023). A direct solution to the above issues is to augment LLMs with external knowledge resources, so as to amend the incorrect generations. Among these resources, structured data (*e.g.,* knowledge graphs and databases), has been widely used as the carrier of the required knowledge for LLMs. Unlike plain text, structured data is organized in a standardized format, conforming to some logical data model. For example, knowledge graphs (KGs) are often organized as fact triples that state the relations between head entities and tail entities, and data tables are organized in the form of column-indexed records by rows. However, as structured data has special data formats or schemas that LLMs have not seen during pre-training, they may be not fully grasped or understood by LLMs (Wei et al., 2021). A straightforward way to solve this problem is to linearize the structured data into a sentence that LLMs can well understand. While, the amount of structured data is often vast, making it infeasible to include all the data records in the input prompt.

Regarding the above challenges, we are inspired by the tool manipulation strategy for augmenting the abilities of LLMs (Schick et al., 2023; Nakano et al., 2021). Our basic idea is to incorporate specialized interfaces (*e.g.,* extracting columns from tables) to manipulate the structured data records. With these interfaces, we can effectively reduce the search space of the data records, and more accurately identify the required evidence to fulfill specific tasks. In this way, LLMs can concentrate on reasoning based on the evidence obtained from the interfaces. To implement the interface-augmented approach, there remain two key problems, namely how to design suitable interfaces for specific tasks and how to utilize them for reasoning by LLMs, which are the focus of this work.

To design suitable interfaces, we regard multiple types of structured data as black-box systems, and design the interfaces to provide accurate, efficient data access and filtering for LLMs. For each

---

[*] Equal contributions.
[†] Corresponding author.

interface, its implementation is dependent on the characteristic of the structured data, while its functionality is general to all LLMs, with just a few arguments for specifying the data requirements. Based on these interfaces, we propose an *Iterative Reading-then-Reasoning (IRR)* framework for LLMs to utilize the interfaces to solve the tasks based on structured data, namely **StructGPT**. This framework considers two major functions to fulfill different tasks, namely collecting relevant evidence (*reading*) and inferring the answer or planning subsequent steps (*reasoning*). Specifically, we propose an *invoking-linearization-generation* procedure to support LLMs in reading and reasoning on the structured data with the help of the external interfaces. By iterating this procedure with provided interfaces, we can gradually approach the target answer to a given question.

To our knowledge, this is the first work that explores how to support LLMs in reasoning on multiple types of structured data (including tables, KGs, and DBs) in a unified paradigm. To evaluate the effectiveness of our approach, we conduct extensive experiments on a wide range of tasks (*e.g.,* KG-based question answering (KGQA), Table-based question answering (TableQA), and DB-based Text-to-SQL). Experimental results on 8 datasets demonstrate that our approach can effectively enhance the reasoning performance of LLMs on structured data in zero-shot and few-shot settings, even comparable with competitive full-data supervised-tuning methods. For example, in KGQA, TableQA, and Text-to-SQL tasks, our approach yields an increase of 11.4% of Hits@1 on WebQSP, 4.2% of accuracy in TabFact, and 4.7% of execution accuracy in Spider respectively, compared to directly using ChatGPT in the zero-shot setting.

## 2   Related Work

**Reasoning over Structured Data.**   Structured data (*e.g.,* knowledge graphs, tables, and databases) is an important knowledge carrier for a variety of QA and reasoning tasks. Early work focuses on designing specific model architectures tailored for each type of structured data, such as graph neural networks (Sun et al., 2018), table Transformers (Herzig et al., 2020), and tree-structured decoder (Wang et al., 2020). While achieving remarkable performance, these approaches lack generality for various types of structured data and are hard to be transferred across different tasks. Re-

cently, with the success of pre-trained language models (PLMs) (*e.g.,* T5 (Raffel et al., 2020), BART (Lewis et al., 2020)), several methods (Raffel et al., 2020; Khashabi et al., 2020) have adopted PLMs as the general encoder or solver for different structured data and tasks. Among them, Unified-SKG (Xie et al., 2022) unifies a number of reasoning tasks over structured data into a text-to-text format, which concatenates the question and the linearized structured data as input, and then fine-tunes T5 to learn to generate the answer. However, UnifiedSKG also requires to tune the model parameters, and is unable to handle large-scale structured data under the limitation of the maximum input length. Instead, our method can utilize the LLM to perform reasoning on structured data without training, and also leverage the interfaces of structured data to better manipulate vast structured data.

**LLMs for Structured Data.**   Benefitting from the strong few-shot and zero-shot capability, recent studies have leveraged LLMs to perform reasoning over structured data (Chen et al., 2023; Li et al., 2023; Cheng et al., 2022; Rajkumar et al., 2022). Existing work can be roughly divided into two types. The first type of method linearizes the structured data into a sentence (*e.g.,* table rows), and feeds it into the LLMs to generate the answer according to in-context exemplars (Cheng et al., 2022; Chen, 2023). For complex questions or structured data, they first decompose it into multiple simple and short ones and then perform linearization and generation (Ye et al., 2023). Another type of method leverages LLMs to evaluate the plausibility of the solution plan based on the knowledge base (Gu et al., 2023), or first generate a solution draft with in-context exemplars and then revise the draft grounding on the knowledge base (Li et al., 2023). However, most of them only focus on a specific type of structured data, and are lack of generality across various data and tasks. In StructGPT, we provide a unified paradigm that is general to various structured data and downstream tasks.

## 3   Preliminary

In this section, we introduce the definition of structured data, which mainly consists of three commonly used types. Then we present the unified problem statement.

**Structured Data**.   Structured data (*e.g.,* data tables and knowledge graphs) refers to the data that

is in a standardized format, conforming to some logical data model (Xie et al., 2022; Chen et al., 2009). Due to the formal structure, it is easy and efficient to access and query structured data using formal languages (*e.g.,* SQL and SPARQL for databases) or specific algorithms (*e.g.,* triples search for knowledge graphs). In this work, we mainly focus on three types of structured data, namely knowledge graphs (KG), data tables (Table), and databases (DB), since they play an important role as the knowledge source in helping solve complex reasoning tasks, described as follows.

• *Knowledge Graph.* A knowledge graph (KG) consists of a number of triples to store the factual knowledge, denoted as $\mathcal{G} = \{\langle e, r, e' \rangle | e, e' \in \mathcal{E}, r \in \mathcal{R}\}$, where $\mathcal{E}$ and $\mathcal{R}$ denote the set of entities and relations, respectively. A triple $\langle e, r, e' \rangle$ represents the fact that there is a relation $r$ between the head entity $e$ and the tail entity $e'$.

• *Data Table.* A data table $\mathcal{T}$ (*table* in short) contains multiple columns $\{c_i\}_{i=1}^{C}$ and rows $\{l_j\}_{j=1}^{R}$, where each row $l_j$ denotes a data record formatted by the attributes indexed by columns $\{c_i\}_{i=1}^{C}$, and $v_{i,j}$ denotes the content in the cell corresponding to the position at column $i$ and row $j$.

• *Database.* A database (DB) typically consists of $N$ data tables, denoted as $\mathcal{D} = \{\mathcal{T}_1, \mathcal{T}_2, ..., \mathcal{T}_N\}$. Besides the column names, the foreign keys across all tables are also available to link the data from two tables, denoted as $\{(c_i^{(k)}, c_j^{(h)})\}$, where $c_i^{(k)}$ and $c_j^{(h)}$ denote the $i$-th and $j$-th columns in the $k$-th and $h$-th tables, respectively.

**Problem Statement**. This work mainly focuses on using LLMs to solve complex reasoning tasks based on structured data. Formally, it can be described as a question answering task: given a natural language question $q$ and an accessible structured data $\mathcal{S}$ (*e.g.,* a knowledge graph, a table, or database), the LLM needs to extract useful evidence from $\mathcal{S}$ and then generates the expected result to answer the question $q$ based on the extracted evidence. According to the task requirement, the generated result can be either free-form answers in natural language or structured expressions (*e.g.,* SQL statements) to be executed for obtaining the answer from $\mathcal{S}$. Since we consider three types of structured data (Section 4), our tasks can be instantiated as follows:

• KG based question answering (KGQA)
• Table based question answering (TableQA)
• DB based semantic parsing (Text-to-SQL)

# 4 Approach

## 4.1 Overview

In this work, we assume that LLMs have to rely on the evidence contained in the structured data to solve the three tasks described in Section 3. An intuitive idea is to conduct a two-stage framework as prior studies on retrieval-augmented approaches (Izacard et al., 2022; Oguz et al., 2022), in which LLMs are employed to first collect sufficient evidence relating to the question and then figure out the answer by the LLMs. However, such an approach is not directly applicable to structured data. Although LLMs are capable of solving diverse tasks in natural language, they have limited capacities in accurately representing and understanding structured data, especially for their contained domain-specific knowledge (Moiseev et al., 2022; Emelin et al., 2022).

To address this difficulty, our solution is inspired by the use of specialized tools in solving complex tasks for LLMs (Nakano et al., 2021; Gao et al., 2022b; Schick et al., 2023). We noted that structured data is well organized and supports easy access via formal language or queries (called *interface* for generality). The basic idea of our approach is to disentangle the two processes of *reading* and *reasoning* for LLMs: we utilize the interface of structure data to implement accurate, efficient data access and filtering (*obtaining the relevant evidence*), and further utilize the reasoning ability of LLMs to figure out the final plan or result for the question (*fulfilling the task*). In this way, LLMs can concentrate on the reasoning process in answering the question, without considering the specialized approach to reading the structure data.

Specially, in our framework, we encapsulate the structure data as a black-box system, and provide specific interfaces for LLMs to access the contained data. Further, we propose an invoking-linearization-generation procedure that enables LLMs to read and extract useful evidence from structured data via the corresponding interface. By iterating the above procedure with provided interfaces, we can gradually obtain the answers by leveraging the superior reasoning abilities of LLMs.

## 4.2 Interfaces for Structured Data

Due to the standardized data formats, structured data is often equipped with efficient data management ways, *e.g.,* SQL for the database. In our approach, we aim to provide LLMs with special-

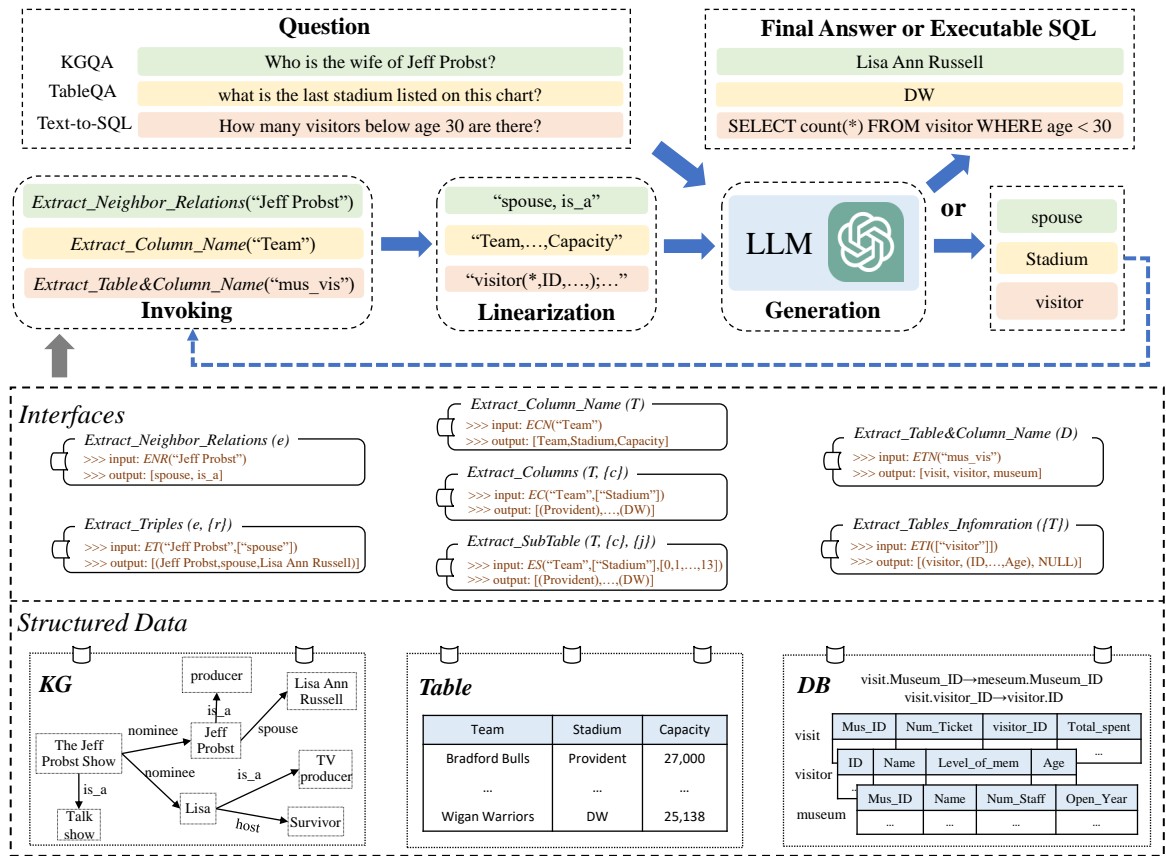

Figure 1: The overview of the proposed iterative reading-then-reasoning approach. We design specialized interfaces for reading structured data, and iterate the invoking-linearization-generation procedure to utilize LLMs for performing reasoning on the interfaces, until deriving the final answer or executable SQL.

ized interfaces, helping LLMs to *read* and *utilize* the structured data. Next, we present the specially designed interfaces for KG, table, and DB.

**Interfaces for Knowledge Graph.** When performing complex reasoning on a KG, existing work (Sun et al., 2018) typically starts from a certain entity (about the question topic), and jumps along with the relations until reaching the answer. In this process, LLMs should be aware of the neighboring relations of the current entity, and the neighboring triples with certain relations to the current entity. Based on it, LLMs can select the relevant relations and triples from them to find the answer. For this purpose, we devise two functions for assisting LLMs to accomplish the above operations.

• *Extract_Neighbor_Relations (e)*: extracts all the neighboring relations of the entity $e$.

• *Extract_Triples (e, {r})*: extracts all the triples with the relation in $\{r\}$ and head entity $e$.

**Interfaces for Table.** Given a data table, LLMs need to know its contained column names, and can access the content by row or column, enabling LLMs to extract its sub-table containing relevant columns and rows. Thus, we define three functions:

• *Extract_Column_Name ($\mathcal{T}$)*: extracts all the column names of a table $\mathcal{T}$.

• *Extract_Columns ($\mathcal{T}$, {c})*: extracts the contents of columns from a table $\mathcal{T}$ by indices $\{c\}$.

• *Extract_SubTable ($\mathcal{T}$, {c}, {j})*: extracts the sub-table specified by the column indices $\{c\}$ and row indices $\{j\}$ from a table $\mathcal{T}$.

**Interfaces for Database.** Considering a simplified setting when querying the database, LLMs should be aware of all the contained tables and columns (by name) for relevant tables selection, and can also acquire the detailed columns and foreign keys from the selected tables to search for the answer. Thus, we devise two functions as follows:

• *Extract_Table&Column_Name ($\mathcal{D}$)*: extracts the names of all the tables and their contained columns from the database.

• *Extract_Tables_Information ({$\mathcal{T}$})*: extracts the table names, column names, and foreign keys from a set of tables $\{\mathcal{T}\}$.

## 4.3 Reading and Reasoning with Interfaces

Based on the above interfaces, we propose a general *invoking-linearization-generation* procedure that can be iterated in multiple turns for utilizing LLMs to perform reading and reasoning on structured data. For each iteration, based on the currently collected data, we first invoke an interface to extract relevant evidence from structure data, then linearize it into a textual prompt, and finally feed the prompt into the LLM for generation (selecting useful data or predicting the answer).

**Invoking an Interface.** In this step, we aim to invoke an interface for extracting the relevant information from the structured data. According to the designed interfaces in Section 4.2, we construct the input based on the currently available data (*e.g.,* entity and table), and then invoke the interface to obtain more detailed relevant information (*e.g.,* neighboring relations and column names), which will be fed into LLMs for collecting useful information or generating the answer.

**Information Linearization.** Given the extracted information, we convert it into a textual sentence that can be understood by LLMs. For the information from KG (*i.e.,* relations and triples), we concatenate them into a long sentence marked by specific separation and boundary symbols. For table and database, we leverage the same way to linearize the extracted table names or column names. While for contents in columns and rows, we follow existing work (Pasupat and Liang, 2015) that first converts them into triples, where head entities are the row indices, relations are column names, and tail entities are the content in the cell, *e.g.,* "*(row 1, year, 1896)*" and "*(row 1, city, Athens)*". Then, for each row, we extract the row indices in the front and omit it in the triples, to compose a simplified sentence, *e.g.,* "*row 1: (year, 1896), (city, Athens)*". For multiple rows, we concatenate them into a long sentence via a special separation symbol.

**LLM for Generation.** After linearization, we design two types of input prompts for LLMs to fulfill different purposes[1]:

• The first type of prompts mostly adopts the following pattern: "*Here are [Y]. Which [X] are most relevant to answer the question [Q]*". It aims to elicit the ability of LLMs to select useful evidence

(*i.e., [X]*) from linearized extracted information (*i.e., [Y]*), according to the question (*i.e., [Q]*).

• The second type of prompt follows the pattern: "*Based on [Y], please generate [Z] for the question [Q]*". It aims to predict the targeted results (*i.e., [Z]*) for the given question (*i.e., [Q]*) based on the linearized extracted information (*i.e., [Y]*). Note that the targeted results can be either the answer string or executable formal language (*e.g.,* SQL) that can lead to the final answer.

By iterating the above invoking-linearization-generation procedure on designed interfaces, LLMs can progressively capture more useful evidence for deriving the final answer.

## 4.4 Instantiated Downstream Tasks

In the following, we describe the instances of the above general workflow for the tasks described in Section 3, since they deal with very different structure data and vary in the task settings.

**KG-based Question Answering (KGQA).** This task aims to find the answer entities for the question based on the KG. Following existing work (Sun et al., 2018), we denote the mentioned entity in the given question $q$ as the topic entity $e_T$, and assume it has been linked to some specific entity on the KG through existing linking tools (*e.g.,* Google Knowledge Graph Search API) or models (*e.g.,* ELQ (Li et al., 2020)). Starting from $e_T$, we perform the invoking-linearization-generation procedure two times using the two interfaces in KG sequentially. First, we invoke the interface *Extract_Neighbor_Relation($e_T$)* to extract the candidate one-hop relations, linearize them to compose the input prompt, and then leverage the LLM to select the useful relations $\{r\}$ according to the question. Then, based on $\{r\}$, we invoke the *Extract_Triples ($e_T$, $\{r\}$)* interface to collect the relevant triples for the head entity $e_T$ and relation in $\{r\}$, then linearize this information, and finally employ the LLM to select the most relevant triples, whose tail entities will be considered as the final answer. Besides, we can also consider the multi-hop KGQA task (Lan et al., 2021), where after selecting the triples of the current hop, the LLM should assess whether the current information is sufficient to answer the question. Then, LLMs will make according actions based on the assessment, *i.e.,* stopping the iterations for producing the answer or continuing the iterations on next-hop tail entities from selected triples.

---

[1]Note that our used prompts are not always consistent with the two examples, as we have rewritten them to better adapt to the specific datasets and extracted information.

**Table-based Question Answering (TableQA).**
For TableQA, we typically need to answer the question according to the content in the given table. We also perform the above procedure by using the three interfaces in turn. Concretely, first, we invoke *Extract_Column_Name ($\mathcal{T}$)* to extract all column names of a table, linearize them, and leverage LLMs to select the relevant ones $\{c\}$ according to the question. Then, we invoke *Extract_Columns ($\mathcal{T}$, $\{c\}$)* to extract the contents of all relevant columns, and select the useful row indices $\{j\}$ by LLMs. Subsequently, we further invoke *Extract_SubTable ($\mathcal{T}$, $\{c\}$, $\{j\}$)* to generate the sub-table for the question. Based on the linearized sub-table, the LLM finally generates the answer to the question.

**DB-based Semantic Parsing (Text-to-SQL).**
This task focuses on generating a SQL query that can be executed to obtain the required information from a database. To achieve this goal, first, we invoke *Extract_Table&Column_Name ($\mathcal{D}$)* to obtain all the table names and their column names in the DB, linearize them, and utilize the LLM to select the relevant table names. Then, we invoke *Extract_Tables_Information ($\{\mathcal{T}\}$)* to obtain all the relevant information (*i.e.,* column names and foreign keys) from these tables. Similarly, by linearizing this information and composing the input prompt, the LLM can generate an executable SQL for the given question.

## 5 Experiment

We conduct experiments on three complex reasoning tasks over structured data, *i.e.,* KGQA, TableQA, and DB based text-to-SQL.

### 5.1 Datasets

For KG based QA (KGQA), we adopt two benchmark datasets, *i.e., WebQuestionsSP* (WebQSP) (Yih et al., 2016) and *MetaQA* (Zhang et al., 2018) for evaluation. The answer entities in WebQSP require up to 2-hop reasoning on the Freebase KG. In contrast, MetaQA contains questions in the movie domain, whose answer entities are up to 3 hops away from the topic entities on a movie KG (based on OMDb). According to the number of hops, it is split into three sub-datasets, *i.e.,* MetaQA-1hop, MetaQA-2hop, and MetaQA-3hop.

For Table based QA (TableQA), we adopt three widely-used datasets, *weakly-supervised WikiSQL* (WikiSQL) (Zhong et al., 2017), *WikiTable-*

*Questions* (WTQ) (Pasupat and Liang, 2015), and *TabFact* (Chen et al., 2020). The first two are typical table-based question answering datasets, and the third one is a multiple-choice dataset that concentrates on table fact verification. WikiSQL requires filtering and aggregating information over the table content, and the WTQ demands more advanced reasoning capabilities (*e.g.,* sorting). TabFact needs to judge whether the provided statement agrees with the facts stored in a table.

For DB based semantic parsing (Text-to-SQL), we adopt three public datasets, *i.e., Spider* (Yu et al., 2018), *Spider-SYN* (Gan et al., 2021), and *Spider-Realistic* (Deng et al., 2021). Spider is a typical Text-to-SQL dataset covering 20 databases with a set of 1034 evaluation samples. Spider-SYN and Spider-Realistic are two more challenging datasets derived from Spider. Concretely, Spider-SYN manually substitutes the synonyms in natural language questions, while Spider-Realistic removes the questions in the evaluation set that explicitly mention the required columns' names.

### 5.2 Evaluation Metrics

For KGQA, we employ Hits@1 which assesses whether the top-1 predicted answer is correct. In our approach, we focus on generating the most confident answer and then checking if the prediction hits any target. As LLMs may generate multiple answers, we also conducted a manual double-check finally (Tan et al., 2023), to judge if wrong answers are included. For TableQA, we adopt two evaluation metrics, namely denotation accuracy and accuracy. In WTQ and WikiSQL, denotation accuracy is employed to evaluate whether the predicted answer is the same as the gold answer based on set-level equivalence. In TabFact, we adopt accuracy to assess the correctness of the prediction. For Text-to-SQL, we adopt the execution accuracy (EX) to assess whether the execution results of the predicted SQL and the gold SQL are the same.

### 5.3 Baselines

We compare our method with competitive full-data supervised-tuning baselines tailored to these tasks. Specifically, our method is a general iterative reading-then-reasoning (IRR) framework that can be used for different LLMs. And we test our IRR with two different LLMs, *i.e.,* Davinci-003 (text-davinci-003 (Ouyang et al., 2022b)) and Chat-

GPT (*i.e.,* gpt-3.5-turbo [2]), under zero-shot and few-shot settings [3]. Considering the evolution of the closed large language model, *e.g.,* ChatGPT, we have further conducted supplementary experiments on three datasets (*i.e.,* WebQSP, WTQ, and Spider) using the latest August version of ChatGPT. The results are presented in Appendix A. For KGQA, we select KV-Mem (Miller et al., 2016), GragtNet (Sun et al., 2018), EmbedKGQA (Saxena et al., 2020), NSM (He et al., 2021), and UniKGQA (Jiang et al., 2023). For TableQA, we select MAPO (Liang et al., 2018), TAPAS (Herzig et al., 2020; Eisenschlos et al., 2020), UnifiedSKG (T5-3B) (Xie et al., 2022), TAPEX (Liu et al., 2022), and DATER (Ye et al., 2023). For Text-to-SQL, we select RAT-SQL+BERT$_{Large}$ (Wang et al., 2020), TKK-Large (Gao et al., 2022a), T5-3B+PICARD (Raffel et al., 2020), RASAT+PICARD (Qi et al., 2022), and RESDSQL-3B+NatSQL (Li et al., 2023).

Additionally, we incorporate baselines that employ Davinci-003 and ChatGPT directly for achieving the aforementioned tasks in a zero-shot setting. To ensure a fair comparison, we utilize the same instruction prompt to evaluate them, ensuring that the only difference with our method is the usage of structured data. Specifically, in KGQA datasets, we follow existing work (Tan et al., 2023) that utilizes LLMs to answer the questions without using KG. In TableQA and Text-to-SQL, we feed the required information of tables with questions into LLMs (Liu et al., 2023c,a), without special treatment for the overlength problem.

## 5.4 Results and Analysis

We show the results on KGQA, TableQA, and Text-to-SQL tasks and analyze them respectively.

**Evaluation on KGQA.** Table 1 shows the results on KGQA datasets. First, LLMs can achieve performance comparable to the supervised learning model (*i.e.,* 61.2 of ChatGPT *v.s.* 66.4 of GraftNet and 48.3 of Davinci-003 *v.s.* 46.7 of KV-Mem) on the WebQSP dataset, in a zero-shot setting without using KGs. It demonstrates that LLMs indeed grasp a certain amount of knowledge that can help them answer complex questions. However, on more difficult datasets that require multi-hop reasoning (*e.g.,* MetaQA-2hop and MetaQA-3hop), the two LLMs perform not well. It indicates that

[2]https://platform.openai.com/docs/models/gpt-3-5
[3]In our experiment, we use the June version of Davinci-003 and ChatGPT.

Table 1: Results of different methods for KGQA (Hits@1 in percent). We copy the results in the first block from He et al. (2021) and Jiang et al. (2023). The best results of each block are highlighted in bold.

| Methods | WQSP | MQA 1hop | MQA 2hop | MQA 3hop |
|---|---|---|---|---|
| KV-Mem | 46.7 | 96.2 | 82.7 | 48.9 |
| GraftNet | 66.4 | 97.0 | 94.8 | 77.7 |
| EmbedKGQA | 66.6 | 97.5 | 98.8 | 94.8 |
| NSM | 68.7 | 97.1 | **99.9** | 98.9 |
| UniKGQA | **75.1** | 97.5 | 99.0 | **99.1** |
| Davinci-003 | 48.3 | 52.1 | 25.3 | 42.5 |
| + IRR (ours) | 71.9 | 94.4 | 59.5 | 70.2 |
| + IRR (ours, few-shot) | 71.0 | 97.1 | 93.5 | 75.3 |
| ChatGPT | 61.2 | 61.9 | 31.0 | 43.2 |
| + IRR (ours) | **72.6** | 94.2 | 93.9 | 80.2 |
| + IRR (ours, few-shot) | 69.6 | **97.1** | **97.3** | **87.0** |

LLMs can not solely rely on their own knowledge to answer difficult questions, and their augmentation with KGs is necessary. In contrast, when incorporating our proposed method to access KG, the performance of Davinci-003 and ChatGPT can be both substantially improved, indicating the effectiveness of our proposed method for supporting LLMs reasoning over KG. By adding a few in-context exemplars (*i.e.,* 15 for WQSP and 32 for MQA) to LLMs, we can further improve the model performance. In our approach, we devise interfaces for KG to efficiently read the relevant information, and leverage LLMs to extract useful parts and perform reasoning. We leverage the IRR procedure on devised interfaces sequentially, which can progressively capture more useful detailed evidence for finally obtaining the answer.

**Evaluation on TableQA.** Table 2 shows the results on three TableQA datasets. First, with the full table as the prompt, ChatGPT can also achieve comparable performance on WTQ and TabFact as full-data supervised-tuning methods, but performs not well on more difficult WikiSQL datasets. It also indicates that LLMs have the capability of understanding the knowledge within table data to some extent. Second, our proposed method can consistently improve the performance of two LLMs a lot in both three datasets. At the same time, when adding 32 in-context exemplars to the LLMs, they can obtain further performance improvements. It indicates the effectiveness of our proposed method in helping LLMs reasoning over Table. Our approach provides a more effective way for LLMs to

Table 2: Results of different methods for TableQA (denotation accuracy for WTQ and WikiSQL, accuracy for TabFact). We copy the results of TAPAS on TabFact from Eisenschlos et al. (2020), and others in the first block from their original papers. The best results of each block are highlighted in bold.

| Methods | WTQ | WikiSQL | TabFact |
|---|---|---|---|
| MAPO | 43.8 | 72.6 | - |
| TAPAS | 48.8 | 83.6 | 81.0 |
| UnifiedSKG (T5-3B) | 49.3 | 86.0 | 83.7 |
| TAPEX | 57.5 | **89.5** | 84.2 |
| DATER | **65.9** | - | **93.0** |
| Davinci-003 | 34.8 | 49.1 | 80.7 |
| + IRR (ours) | 39.2 | 51.8 | 76.5 |
| + IRR (ours, few-shot) | **57.0** | 64.6 | 87.3 |
| ChatGPT | 43.3 | 51.6 | 82.9 |
| + IRR (ours) | 48.4 | 54.4 | 87.1 |
| + IRR (ours, few-shot) | 52.2 | **65.6** | **87.6** |

Table 3: Performance comparison of different methods for Text-to-SQL (execution accuracy in percent). We copy the results of RAT-SQL+BERT$_{Large}$ and TKK-Large from Deng et al. (2021) and Gao et al. (2022a), respectively. And we copy the results of the other three methods in the first block from Liu et al. (2023b). The best results of each block are highlighted in bold.

| Methods | Spider | Spider-SYN | Spider-Realistic |
|---|---|---|---|
| RAT-SQL + BERT$_{Large}$ | 72.3 | - | 62.1 |
| TKK-Large | 73.2 | 60.5 | 64.4 |
| T5-3B + PICARD | 79.3 | 69.8 | 71.4 |
| RASAT + PICARD | 80.5 | 70.7 | 71.9 |
| RESDSQL-3B + NatSQL | **84.1** | **76.9** | **81.9** |
| Davinci-003 | 68.8 | 60.1 | 63.2 |
| + IRR (ours) | 69.5 | 60.3 | 64.2 |
| + IRR (ours, few-shot) | 72.7 | 63.2 | 70.7 |
| ChatGPT | 70.1 | 58.6 | 63.4 |
| + IRR (ours) | 74.8 | 62.0 | 70.3 |
| + IRR (ours, few-shot) | **77.8** | **64.0** | **72.0** |

iteratively access and utilize the relevant information from the table, which reduces the influence of irrelevant and redundant information.

**Evaluation on Text-to-SQL.** Table 3 shows the results on DB-based datasets. First, with all the information from DB (table names, column names, and foreign keys) as the prompt, the LLMs have the capability of directly generating a suitable SQL query of the question, performing well on all three datasets. Whereas, the performance of LLMs is not better than competitive full-data supervised-tuning methods, showing the difficulty of this task. As our proposed method can extract relevant tables and columns, it also alleviates the influence of irrelevant information for LLMs to generate the SQL query. Simultaneously, with the assistance of 32 in-context exemplars, LLMs exhibit enhanced comprehension of the mapping between natural language questions and their corresponding SQL queries. The consistent performance improvements over the three datasets whenever in zero-shot or few-shot settings also indicate the effectiveness of our proposed method.

**Case Study.** We show an example of KGQA in Figure 2, to help understand the working process of our method. Given the question, the interfaces of the structured data are sequentially invoked to iteratively extract more useful and detailed information. In each iteration, we first invoke the Extract_Neighbor_Relations function to extract the neighboring relations (*e.g., birthplace, residence, and education*) of the topic entity "*Harper Lee*",

then linearize them and compose the input prompt. Here, we utilize the instruction (*i.e.,* provide only one relevant relation that's present in the candidate) to elicit the LLM to generate the most relevant relation, *i.e., education*. Based on the selected relation, we further invoke the Extract_Triples function to extract the triples with the relation to the topic entity. After linearization, another instruction (*i.e.,* you just need to provide only one answer entity), is adopted for guiding the LLM to generate the final answer, *i.e., Monroe County High School*. Besides, we show the representative examples of TableQA and Text-to-SQL in Appendix B.

**Error Analysis.** To systemically analyze the shortcomings of our approach, we first select three datasets (*i.e.,* WebQSP, WTQ, and Spider) with different types of structured data, and randomly sample 100 error cases from each dataset. Then, we manually examine these failures and classify them into five categories:

• **Selection Error**: the relevant information has not been selected by the LLM.

• **Reasoning Error**: given the extracted relevant information, the LLM fails to generate the ground-truth answer or SQL.

• **Generation Format Error**: the generated answer is in an abnormal format that fails to be identified by our result parser.

• **Hallucination**: the generated results are inconsistent with the extracted information.

• **Other Errors**: other uncategorizable errors.

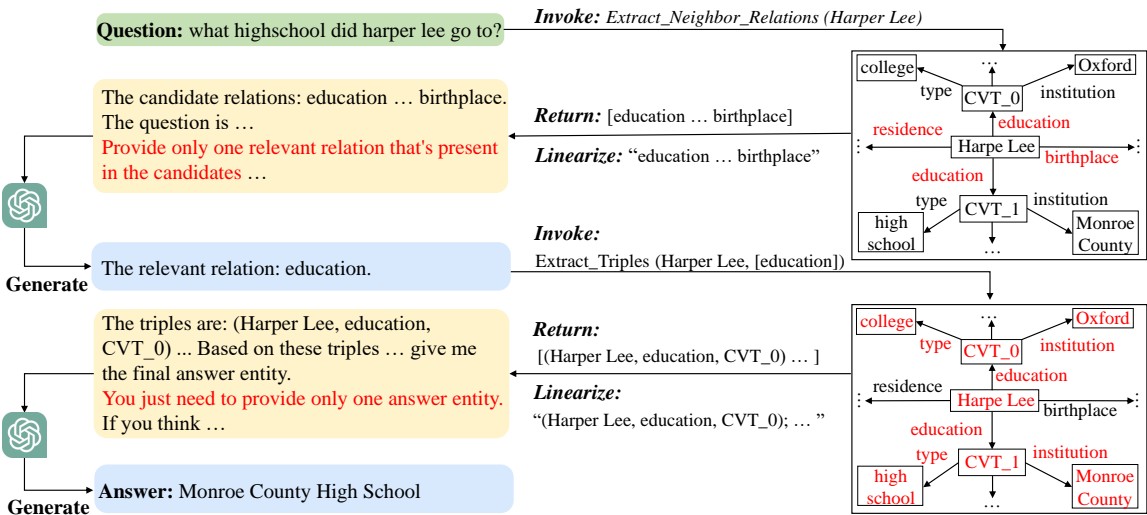

Figure 2: Case study of our method on WebQSP.

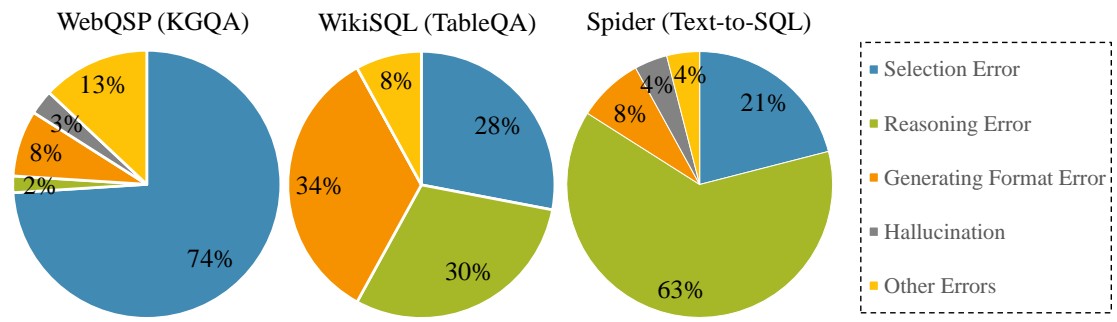

Figure 3: Proportions of different error types in three datasets over different types of structured data.

We show the statistics in Figure 3. First, for the three datasets, the distributions of occurring errors are different. In WikiSQL, the frequencies of generation format, selection, and reasoning errors are relatively uniform. Whereas, in WebQSP, the selection error is the major error type (74%), since the KGQA task requires selecting the most relevant one from thousands of relations, which is not easy work. In Spider, reasoning error occurs more (62%), since the Text-to-SQL task requires LLMs to generate a SQL that can be executed to obtain the answer, which is also hard for LLMs.

According to the error distributions, it is promising to refine the major error cases to specifically improve the performance on each dataset. Concretely, we can devise more high-quality prompts that elicit LLMs to carefully make decisions when selecting and reasoning on KGQA and Text-to-SQL tasks, respectively. Besides, we also consider adding more interfaces and iteration turns for decomposing the hard probelm into multiple simple ones, to simplify the complex reasoning task for better performance. We will try the above solutions in our future work.

## 6 Conclusion

In this work, we proposed a general framework for improving the zero-shot reasoning ability of LLMs over structured data, namely StructGPT. In our approach, we first constructed the specialized interfaces that support accurate and efficient data access, and then proposed an invoking-linearization-generation procedure that leverages LLMs to read and perform reasoning based on the interface. By iterating the above procedure using the interfaces sequentially, LLMs can progressively capture more useful and detailed evidence and finally generate the answer. To verify the effectiveness of our approach, we implemented our approach on KG based QA, table based QA and DB based semantic parsing tasks. Experimental results on 8 datasets show that our approach can boost the zero-shot performance of LLMs by a large margin, and achieve comparable performance as full-data supervised-tuning methods. We also provide detailed error analysis to point out the weakness of our approach, for enlighting other researchers in related areas.

# 7 Limitations

Although StructGPT demonstrates remarkable performance across tasks over structured data, there are some limitations of our method. First, the two LLMs used in our model, *i.e.,* ChatGPT and Davinci-003, have a strong capability of following instructions. Hence, more experiments are required to evaluate our method with in-context learning on other LLMs that perform poorly at instruction following. Similarly, we only evaluate question answering tasks based on structured data. Future work should include wider evaluation scenarios to evaluate the universality of our method, *e.g.,* data-to-text and formal-language-to-text (Xie et al., 2022). Finally, since it is difficult to control the answer format during the generation process of LLMs in different datasets, there are several format errors in generated texts as shown in Section 5. Therefore, the performance of our method can be further improved by meticulously designing the prompt and answer parsing for different datasets.

## Acknowledgments

This work was partially supported by National Natural Science Foundation of China under Grant No. 62222215, Beijing Natural Science Foundation under Grant No. 4222027 and L233008. And this work is also partially supported by the Outstanding Innovative Talents Cultivation Funded Programs 2022 of Renmin University of China. Xin Zhao is the corresponding author.

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

Table 4: Results of different version of ChatGPT for WebQSP, WTQ, and Spider.

| Methods | WebQSP | WTQ | Spider |
|---|---|---|---|
| ChatGPT (June) | 61.2 | 43.3 | 70.1 |
| ChatGPT (June) + IRR | 72.6 | 48.4 | 74.8 |
| ChatGPT (August) | 62.1 | 41.1 | 75.2 |
| ChatGPT (August) + IRR | 75.3 | 50.4 | 77.1 |

## A Experiment With Latest Version of LLM

We have noted that the ChatGPT is continuously evolving. Furthermore, we have conducted supplementary experiments on three datasets using the latest August version of LLM. The results are presented in the Table 4. It is noteworthy that Chat-GPT indeed continuously evolves, as evidenced by its distinct performance compared to that of the June version. Although the evolved ChatGPT underperforms compared to the June version on the WTQ dataset, our approach can consistently further enhances the ChatGPT performance with the evolved version on all three tasks. It indicates the robustness of our proposed method.

## B Case Study

Here, we select one representative example for each type of structured data and present the case study in Figure 4. For KG, we first invoke the Extract_Neighbor_Relations function to extract the neighboring relations (*e.g., birthplace, residence, and education*) of the topic entity "*Harper Lee*", then linearize them and compose the input prompt. In the prompt, we utilize the instruction (*i.e.,* provide only one relevant relation that's present in the candidate) to elicit the LLM to generate the most relevant relation, *i.e., education*. Based on the selected relation, we further invoke the Extract_Triples function to extract the triples with the relation to the topic entity. After linearization, another instruction (*i.e.,* you just need to provide only one answer entity), is adopted for guiding the LLM to generate the final answer, *i.e., Monroe County High School*.

For table, we first invoke the Extract_Column_Name function to extract the column names from the table for linearization, and then design the prompt (*i.e.,* which columns are most relevant to answering the question?) for the LLM to select the useful columns, *i.e., District and Incumbent*. Then, by using the Extract_Columns and Extract_SubTable functions and proper instructions, we elicit the LLM to select the useful row indices (*i.e.,* item 8) and finally generate the answer (*i.e.,* 19th).

For database, we also first invoke the Extract_Table&Column_Name to extract all the table names and column names, linearize them and utilize the instruction (*i.e.,* which tables do you need to complete the SQLite SQL query?) to prompt the LLM. Then, based on the selected tables (*i.e.,* Dogs and Breeds), we further invoke the Extract_Tables_Information function and prompt the LLM via an instruction (*i.e.,* complete sqlite SQL query only with no explanation) to generate the SQL for the question, which can be executed to obtain the final answer.

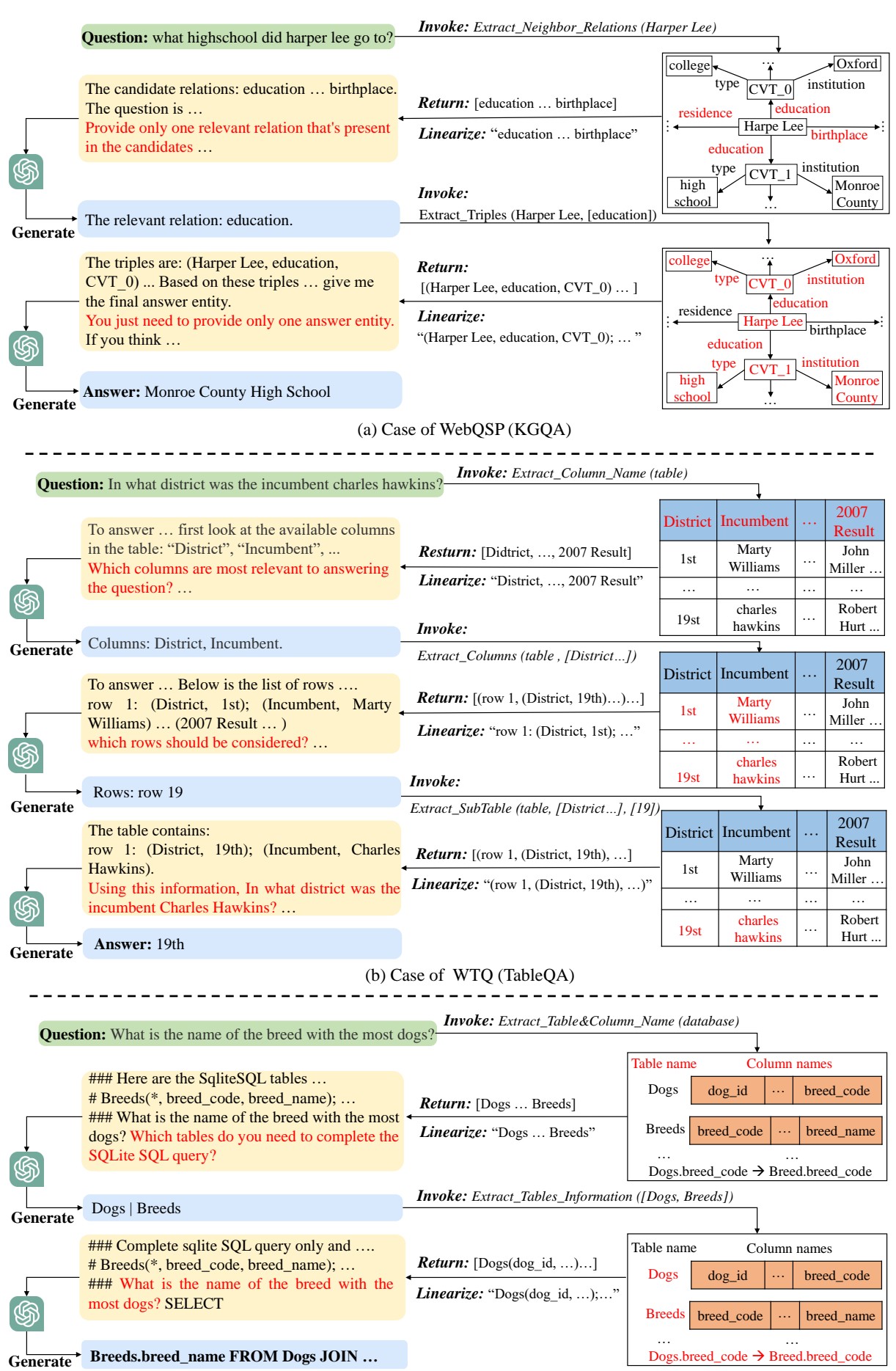

Figure 4: Case study of our method on KGQA, TableQA and Text-to-SQL task.