# OpenReview forum: "StructGPT: A General Framework for Large Language Model to Reason over Structured Data"
_EMNLP/2023/Conference — EMNLP 2023 Main_

### Official Review · Reviewer_KDgm · 2023-08-04

**Soundness:** 3

**Excitement:**

3: Ambivalent: It has merits (e.g., it reports state-of-the-art results, the idea is nice), but there are key weaknesses (e.g., it describes incremental work), and it can significantly benefit from another round of revision. However, I won't object to accepting it if my co-reviewers champion it.

**Paper Topic And Main Contributions:**

The paper proposes an Iterative Reading-then-Reasoning (IRR) framework for LLMs utilizing interfaces to structured data. The framework considers two major functions to fulfill two different tasks, namely collecting relevant evidence (i.e. *reading*) and inferring the answer or planning subsequent steps (i.e., *reasoning*). The paper proposes an invoking-linearization-generation procedure to support LLMs in reading and reasoning on structured data with the help of external interfaces. By iteratively applying this procedure with different interfaces, the target answer can be gradually approached to a given question.

The authors conduct experiments on a wide range of tasks (e.g., KG-based question answering, Table-based question answering, and DB-based Text-to-SQL) and demonstrates that the IRR framework can effectively enhance the reasoning performance of LLMs on structured data in zero-shot and few-shot settings but not compared to prior art systems that use  supervised-tuning methods.

**Questions For The Authors:**

Why does the LLM extract spouse, Stadium or visitor, as indicated in figure 1?
- how do you extract for example the name Harper Lee from the question? Is a named entity extraction tool used? Or is this another call to an LLM?(figure 2)

**Reasons To Accept:**

- novel combination of KG, table and text-to-SQL approaches utilizing LLMs

**Reasons To Reject:**

unclear description of approach
	- Figure 1 is quite confusing with multiple arrows pointing to an LLM box
	- It's unlcear how the prompts interact with the interface. Is this an agent-based approach?
- performance is below or just en par with prior art work
- Reproducibility with chatGPT is difficult due to non-determinism of the system and it is unclear with model version was used [addressed by rebuttal]

**Reproducibility:**

3: Could reproduce the results with some difficulty. The settings of parameters are underspecified or subjectively determined; the training/evaluation data are not widely available.

**Reviewer Confidence:**

3: Pretty sure, but there's a chance I missed something. Although I have a good feel for this area in general, I did not carefully check the paper's details, e.g., the math, experimental design, or novelty.

---

> ### Author Rebuttal · Authors · 2023-08-29
>
> Thanks for your insightful suggestions and we have listed our response to your concerns as follows.
>
> **1.unclear description of approach - Figure 1 is quite confusing with multiple arrows pointing to an LLM box - It's unclear how the prompts interact with the interface. Is this an agent-based approach?**
>
> The core process of our approach is the iterative invoking-linearization-generation procedure illustrated in Figure 1. Concretely, at each iteration, we first invoke the interface to extract relevant information from structured data. Secondly, we convert the extracted information into a textual sentence by linearizing it. Next, we construct the LLMs' input by combining the linearized extracted information **(arrow from "Linearization" box to "LLM" box)** and the question **(arrow from "Question" box to "LLM" box)**. Finally, the LLMs select useful evidence for next iteration of reasoning **(arrow from "LLM" box to selected information box)** or generating the target outputs **(arrow from "LLM" box to "Final Answer or Executable SQL" box)**. In Figure 1, we present the data flow of the three types of tasks in parallel, distinguished by different colors. The solid arrows indicate the data flow, while the dashed arrows represent multiple iterations. Here, we further present an example of KGQA task to explicate the aforementioned process along with the prompts utilized in each step.
>
> ```
> Invoking:
> Extract_Neighbor_Relations (Harper Lee)
>
> Return:
> [fictional_universe.fictional_character_creator.fictional_characters_created, en, ... ,award.award_nominee.award_nominations, award.award_winner.awards_won, base.biblioness.kjauthors]
> ------------------------------------------
> Linearization:
> fictional_universe.fictional_character_creator.fictional_characters_created, en, ... ,award.award_nominee.award_nominations, award.award_winner.awards_won, base.biblioness.kjauthors.
> ------------------------------------------
> Generation:
> The candidate relations: fictional_universe.fictional_character_creator.fictional_characters_created, en, ... ,award.award_nominee.award_nominations, award.award_winner.awards_won, base.biblioness.kjauthors.
> The question is "what highschool did harper lee go to?" and you'll start with "harper lee". To answer this question, typically you would need to identify some relations that correspond to the meaning of the question. Therefore, select one relation from the candidate relations above that can be used to answer the question. Provide only one relevant relation that's present in the candidates, and begin your response with "The relevant relation:
>
> Output:
> The relevant relation: people.person.education
> ------------------------------------------
> Invoking:
> Extract_Triples (Harper Lee, [people.person.education])
>
> Return:
> [[Harper Lee, people.person.education, CVT_0], [CVT_0, education.education.end_date, 1944], [CVT_0, education.education.institution, Monroe County High School]...]
> ------------------------------------------
> Linearization:
> (Harper Lee, people.person.education, CVT_0), (CVT_0, education.education.end_date, 1944), (CVT_0, education.education.institution, Monroe County High School).
> ------------------------------------------
> Generation:
> The triples are: (Harper Lee, people.person.education, CVT_0), (CVT_0, education.education.end_date, 1944), (CVT_0, education.education.institution, Monroe County High School).
> Based on these triples, if you believe you have gathered sufficient information to answer "what highschool did harper lee go to?", give me the final answer entity and start your response with "The final answers:". You just need to provide only one answer entity. If you think you still do not have enough information to answer the question, respond "Need further information".
>
> Output:
> The final answer: Harper Lee went to Monroe County High School.
> ```
>
>
> As you can see, such a process indeed enable the LLMs as an agent to interact with various structured data under a general framework, which is not be easily implemented by traditional agent-based approaches.
>
> **2.performance is below or just on pair with prior art work**
>
> It should  be noted that prior SOTA methods are almost small models  that are specially fine-tuned by  the training set in the experiments. In addition, a model trained on some dataset is difficult to generalize or work well on another different evaluation dataset. As  a comparison, our approach are zero-shot or few-shot learning, and a fixed LLM solve all  types of tasks. Although LLMs have shown surprising performance recently, they still difficultly handle the structured data without any fine-tuning, whose performance has a significant gap compared to existing supervised fine-tuned models. Our approach helps LLM greatly reduce this performance gap in three typically structured data, and even surpass partial supervised fine-tuned models, which demonstrates the superiority and generality of our approach. Performance improvement is just to illustrate the effectiveness of our approach. Most importantly, our work provide an new perspective how to apply the LLMs to the structured data, which is meaningful to deploy LLM in realistic scenarios.
>
> **3.Reproducibility with chatGPT is difficult due to non-determinism of the system and it is unclear with model version was used**
>
> The difficult reproducibility problem of ChatGPT mainly coms from the decoding randomness and the continuously evolved model version. Both of them are objective factors that can not  be controlled by our method and we have try our best to mitigate their effects.
>
> First, during our experiment, we set all the hyper-parameters relevant to decoding randomness (e.g., temperature) as the value of no randomness (e.g., 0) according  to the official guidelines of OpenAI. And we have open sourced all of the used prompt, hyper-parameter config, and the final ChatGPT output to ensure the reproducibility of our results. And the June version of ChatGPT was utilized in the experiments, and our results can be fully reproduced under this version. We will supplement all these details in the revised paper.
>
> Second, our approach is a general LLM-agnostic framework, whose effectiveness would not be influenced by the different model versions. We conduct the supplementary experiments on three datasets using the latest August version of LLM. The results are presented in the following table. It is noteworthy that ChatGPT indeed continuously evolves, as evidenced by its distinct performance compared to that of the June version. Although the evolved ChatGPT underperforms compared to the June version on the WTQ dataset, our approach can consistently further enhances the ChatGPT performance with the evolved version on all three tasks. It indicates the robustness of our proposed method.
>
> \\begin{array} {|l|c|c|c|}
> \\hline
> \\textrm{Model} & \\textrm{WebQSP} & \\textrm{WTQ} & \\textrm{Spider} \\\\
> \\hline
> \\textrm{ChatGPT (June)} & 61.2 & 43.3 & 70.1 \\\\
> \\hline
> \\textrm{ChatGPT (August)} & 62.1 & 41.1 & 75.2 \\\\
> \\hline
> \\textrm{+ IRR (June)} & 72.6 & 48.4 & 74.8 \\\\
> \\hline
> \\textrm{+ IRR (August)} & 75.3 & 50.4 & 77.1 \\\\
> \\hline
> \\end{array}
>
>
> **4.Why does the LLM extract spouse, Stadium or visitor, as indicated in figure 1? how do you extract for example the name Harper Lee from the question? Is a named entity extraction tool used? Or is this another call to an LLM?(figure 2)**
>
> In Figure 1, the output from the LLMs indicates that they have selected the required information from the linearized input data, which was extracted by invoking the interfaces of structured data. In detail, the "spouse" is the selected relevant **relation** after invoking the *Extract_Neighbor_Relations()* interface of KG, the "Stadium" is the selected relevant **column** name after invoking the *Extract_Column_Name()* interface of table, and the "visitor" is the selected relevant **table** after invoking *Extract_Table&Column_Name()* interface of DB. You can refer to the above answer to Q1 for the detail description of our approach process.
>
> As for the extracted "Harper Lee" from the question, it can be extracted via entity linking. As described in Section 4.4, for KGQA dataset, we just follow existing work [1-4] to perform entity linking in advance with existing linking tools (e.g., Google Knowledge Graph Search API) and models (e.g., ELQ [5])  for performing entity linking instead of using golden annotated ones. Please note, we don't assume directly given the golden entity linking results to sure that our approach can be applied to realistic scenarios. Our experiment settings are consistent with existing works on KGQA tasks, which  makes sure a fair comparison with baselines in Table 1.
>
> Reference:
>
> [1] Haitian Sun, Bhuwan Dhingra, Manzil Zaheer, Kathryn Mazaitis, Ruslan Salakhutdinov, William W. Cohen: Open Domain Question Answering Using Early Fusion of Knowledge Bases and Text. EMNLP 2018.
>
> [2] Apoorv Saxena, Aditay Tripathi, Partha P. Talukdar: Improving Multi-hop Question Answering over Knowledge Graphs using Knowledge Base Embeddings. ACL 2020.
>
> [3] Gaole He, Yunshi Lan, Jing Jiang, Wayne Xin Zhao, Ji-Rong Wen: Improving Multi-hop Knowledge Base Question Answering by Learning Intermediate Supervision Signals. WSDM 2021.
>
> [4] Jiaxin Shi, Shulin Cao, Lei Hou, Juanzi Li, Hanwang Zhang: TransferNet: An Effective and Transparent Framework for Multi-hop Question Answering over Relation Graph. EMNLP 2021.
>
> [5] Belinda Z. Li, Sewon Min, Srinivasan Iyer, Yashar Mehdad, Wen-tau Yih: Efficient One-Pass End-to-End Entity Linking for Questions. EMNLP 2020.

---

### Official Review · Reviewer_1ymw · 2023-08-05

**Soundness:** 5

**Excitement:**

3: Ambivalent: It has merits (e.g., it reports state-of-the-art results, the idea is nice), but there are key weaknesses (e.g., it describes incremental work), and it can significantly benefit from another round of revision. However, I won't object to accepting it if my co-reviewers champion it.

**Paper Topic And Main Contributions:**

The paper presents a framework called Iterative Reading-then-Reasoning (IRR) that enhances the inference capability of LLM in specific tasks by incorporating structured knowledge obtained through the pre-designed external interfaces. The structured knowledge includes three aspects - Knowledge, Table, and Database. The authors design interfaces and implement the entire framework for KGQA, TableQA, and Text-to-SQL tasks. They conducted experiments on multiple typical datasets for these tasks and the results demonstrate that the framework outperforms ChatGPT zero-shot significantly.

**Reasons To Accept:**

- The article has a logical and well-structured writing style.
- Considering the outstanding performance of LLMs in various NLP tasks, the motivation behind the paper is reasonable. Investigating how to enhance the effectiveness of LLMs in domain-specific tasks that require domain knowledge is a meaningful research topic.
- The article provides rich experimental results and case analyses.

**Reasons To Reject:**

As mentioned above, the motivation of this paper is reasonable. However, from a research perspective, this paper represents a relatively lightweight work. For the community, the significance of this paper lies more in validating the ability of ChatGPT to integrate specific knowledge. My concerns regarding this paper are as follows:

- The actual content of the paper is rather limited, and there are instances of redundant descriptions. For example, the two paragraphs from line 229 to line 272 redundantly describe the motivation and work approach already covered in the introduction. Additionally, I suggest that since the author has mentioned the lack of domain knowledge in LLMs for specific tasks, it would be helpful to present a small-scale validation experiment using any of the datasets mentioned in this paper.
- Since ChatGPT has been used as an LLM, it is recommended to mention the date of its retrieval, as different versions of the model may yield different results.
- There are already existing open-source toolkits that integrate a multitude of tool calls (e.g., langchain), and the author should provide some descriptions or examples to illustrate the characteristics and advantages of the proposed method compared to these existing tools.

**Reproducibility:**

5: Could easily reproduce the results.

**Reviewer Confidence:**

5: Positive that my evaluation is correct. I read the paper very carefully and I am very familiar with related work.

---

> ### Author Rebuttal · Authors · 2023-08-29
>
> Thanks for your insightful suggestions and we have listed our response to your concerns as follows.
>
> **1.The actual content of the paper is rather limited, and there are instances of redundant descriptions. For example, the two paragraphs from line 229 to line 272 redundantly describe the motivation and work approach already covered in the introduction. Additionally, I suggest that since the author has mentioned the lack of domain knowledge in LLMs for specific tasks, it would be helpful to present a small-scale validation experiment using any of the datasets mentioned in this paper.**
>
> Thanks for your valuable suggestion. In the approach section, we provide a comprehensive overview that aims to ensure this part is self-contained and easily comprehensible. We will revise this section by providing a more concise overview and adding the detailed prompt designs and interaction ways for better illustrating our method after the review period.
>
> In fact, in our experiment, we have reported the performance of LLMs for specific tasks. For the example of KGQA task, the WebQSP and MetaQA datasets need world knowledge and movie domain knowledge, respectively. We directly utilize LLMs to answer questions without accessing any external knowledge source and the results are presented in Table 1. As we can see, LLMs perform poorly on these datasets. In contrast, their performance significantly improves when leveraging the domain knowledge from KG with our proposed method. This highlights the lack of domain-specific knowledge in LLMs for specific tasks and demonstrates how our approach effectively mitigates this issue by enabling access to external structured data. We will also add this description to illustrate the lack of domain knowledge in LLMs for specific tasks in our paper following your advice.
>
> **2.Since ChatGPT has been used as an LLM, it is recommended to mention the date of its retrieval, as different versions of the model may yield different results.**
>
> Thanks for your suggestion. We will note that the June version of ChatGPT was utilized in the experiment section. Furthermore, we have conducted supplementary experiments on three datasets using the latest August version of LLM. The results are presented in the following table. It is noteworthy that ChatGPT indeed continuously evolves, as evidenced by its distinct performance compared to that of the June version. Although the evolved ChatGPT underperforms compared to the June version on the WTQ dataset, our approach can consistently further enhances the ChatGPT performance with the evolved version on all three tasks. It indicates the robustness of our proposed method.
>
> \\begin{array} {|l|c|c|c|}
> \\hline
> \\textrm{Model} & \\textrm{WebQSP} & \\textrm{WTQ} & \\textrm{Spider} \\\\
> \\hline
> \\textrm{ChatGPT (June)} & 61.2 & 43.3 & 70.1 \\\\
> \\hline
> \\textrm{ChatGPT (August)} & 62.1 & 41.1 & 75.2 \\\\
> \\hline
> \\textrm{+ IRR (June)} & 72.6 & 48.4 & 74.8 \\\\
> \\hline
> \\textrm{+ IRR (August)} & 75.3 & 50.4 & 77.1 \\\\
> \\hline
> \\end{array}
>
> **3.There are already existing open-source toolkits that integrate a multitude of tool calls (e.g., langchain), and the author should provide some descriptions or examples to illustrate the characteristics and advantages of the proposed method compared to these existing tools.**
>
> Thanks for your suggestion and we will add this discussion in our next version paper following your advice. Existing open-source toolkit such as langchain is a Python development framework that allows developers to expand the capabilities of LLMs, and build end-to-end language model applications. Under this framework, LLMs can access external tools, such as search engines, math calculators, and Python interpreters, to easily expand their capabilities. Simultaneously, it enables LLMs to access external data resources using index and retrieval tools. However, the framework has two major limitations that make it difficult to process structured data.
>
> Firstly, the **data format of structured data** is different from normal document format, which makes it hard for LLMs to understand it using traditional dense retrieval tools which ignores the structure information.
>
> Secondly, the framework's **direct retrieval then utilization** approach has limited ability to support LLMs in solving complex user queries that requires multi-hop reasoning over structured data.
>
> In contrast, our proposed iterative reading-then-reasoning approach allows LLMs to **iteratively access external structured data** and **select the required data** to perform reasoning using designed interfaces of structured data and invoking-linearization-generation procedures. Besides, our approach is scalability and flexibility, as we just require to design new interfaces specifically for new structured data and tasks in our framework.

---

### Official Review · Reviewer_b2MA · 2023-08-05

**Soundness:** 5

**Excitement:**

3: Ambivalent: It has merits (e.g., it reports state-of-the-art results, the idea is nice), but there are key weaknesses (e.g., it describes incremental work), and it can significantly benefit from another round of revision. However, I won't object to accepting it if my co-reviewers champion it.

**Missing References:**

The idea of having an intermedia functions/interfaces is similar to designing an intermedia query language (e.g., "Towards Complex Text-to-SQL in Cross-Domain Database with Intermediate Representation").

**Paper Topic And Main Contributions:**

This paper developed an iterative reading-then-reasoning framework for tasks that involve querying structured data (e.g., tables, knowledge graph or relational databases). The key idea is to decompose the mapping from input NL to output answer/SQL into a series of reading and reasoning steps. The reading step is based on some specialized APIs that make it easier to retrieve relevant columns/relations of the entities mentioned in NL. Then during the reasoning step, models can select and use necessary columns/relations to synthesize a SQL or directly generate an answer. The design of intermedia data-accessing functions are shown to be very useful to augment LLM for handling structured data such as tables, knowledge graphs and relational database.

**Reasons To Accept:**

* a simple and universal method for helping LLMs handle structured data
* the experiments and error analysis are informative and thorough

**Reasons To Reject:**

* The experiments can be more convincing. As far as I understand, the current method needs to assume the entities are extracted and given so that the intermedia functions/interfaces can be utilized. This is a strong assumption: 1) entity extraction and linking is not trivial, 2) it breaks the end-to-end property of semantic parsers.  Discussions (or even experiments) should be included.

**Reproducibility:**

4: Could mostly reproduce the results, but there may be some variation because of sample variance or minor variations in their interpretation of the protocol or method.

**Reviewer Confidence:**

3: Pretty sure, but there's a chance I missed something. Although I have a good feel for this area in general, I did not carefully check the paper's details, e.g., the math, experimental design, or novelty.

---

> ### Author Rebuttal · Authors · 2023-08-29
>
> Thanks for your insightful suggestions and we have listed our response to your concerns as follows.
>
> **1.The experiments can be more convincing. As far as I understand, the current method needs to assume the entities are extracted and given so that the intermedia functions/interfaces can be utilized. This is a strong assumption: 1) entity extraction and linking is not trivial, 2) it breaks the end-to-end property of semantic parsers. Discussions (or even experiments) should be included.**
>
> Thank you for your valuable feedback. Here, we want to first outline the task formulations for three types of tasks as the following table.
> \\begin{array} {|c|c|c|c|}
> \\hline
> \\textrm{Task} & \\textrm{input} & \\textrm{Structured~Data} & \\textrm{Output} \\\\
> \\hline
> \\textrm{KGQA} & \\textrm{question} &  \\textrm{KG} &  \\textrm{answer entities}  \\\\
> \\hline
> \\textrm{TableQA} & \\textrm{question} &  \\textrm{Table} &  \\textrm{answers}  \\\\
> \\hline
> \\textrm{Text-to-SQL} & \\textrm{question} &  \\textrm{DB} &  \\textrm{SQL}  \\\\
> \\hline
> \\end{array}
>
> (1) Only for  KGQA task, entity linking is essential, which indeed is a **standard and necessary preprocessing** operation to support subsequent reasoning process [1-4]. Notably, to ensure the practical application, existing work [1-4] adopts available tools (e.g., Google Knowledge Graph Search API) and models (e.g., ELQ [5])  for performing entity linking instead of using golden annotated ones. We just follow such preprocessing paradigm and focus more on the afterward reasoning process. Actually, it is very expensive to implement entity linking through LLM, as the KG generally consists of millions of entities. As a comparison, the traditional entity linking tools and models work more efficiently and can even achieve better performance. In this way, our experimental settings are **consistent with existing KGQA work**, which ensures fair comparisons with baselines presented in Table 1. In the future work, we would investigate the potential of utilizing LLMs to implement entity linking.
>
> (2) For TableQA and Text-to-SQL tasks, our task formulation also aligns with previous supervised fine-tuned models, where the input consists of the question with table information (for TableQA) or DB information (for Text-to-SQL), and the output is the answers (for TableQA) or SQL (for Text-to-SQL). Differently, our approach focuses on the zero-shot settings without any fine-tuning on these tasks, hence we can not rely on supervised tuning (used in previous work) to adapt into the special structured data format. And our experiments in Table 2 and Table 3 (ChatGPT v.s. ChatGPT + IRR) have shown that simply feeding the complete structured data information (e.g., table or DB) into the LLM can not achieve a good performance. Therefore, we propose a unified framework that separately fulfills all the sub-tasks within the end-to-end reasoning process. Concretely, LLMs first select partially required information from the original structured data (e.g., the sub-table content for TableQA or the DB meta-information for Text-to-SQL), which is then utilized to generate target outputs. Such a way can reduce the input length, mitigate the effect of noisy information, and facilitate zero-shot generalization using our designed prompts for improved performance.
>
> **2.The idea of having an intermedia functions/interfaces is similar to designing an intermedia query language (e.g., "Towards Complex Text-to-SQL in Cross-Domain Database with Intermediate Representation").**
>
> Thanks for your recommendation, and we will add this reference. In this work, we aim to propose a unified and general framework that supports LLMs for reasoning over various structured data. In this way, our approach is not binded to downstream specific tasks, and can solve them only needs the corresponding interfaces. Instead, due to the diverse formats of structured data, it is challenging to employ a proper custom query language to unify them.
>
> Concretely, we propose to design the intermediate interfaces tailored to each type of structured data and establish a unified interaction approach between LLMs and these interfaces. This design ensures the scalability and flexibility of our approach, as we just require to design new interfaces specially for new structured data and tasks in our framework. Your recommended work has provided us with new insights, prompting us to consider transforming questions into an intermediate representation that bridges natural language and the final Query Language. We consider to try this idea in the future, and we also welcome further discussion on developing a new programming language for enhancing interaction between LLM and structured data.
>
> Reference:
>
> [1] Haitian Sun, Bhuwan Dhingra, Manzil Zaheer, Kathryn Mazaitis, Ruslan Salakhutdinov, William W. Cohen: Open Domain Question Answering Using Early Fusion of Knowledge Bases and Text. EMNLP 2018.
>
> [2] Apoorv Saxena, Aditay Tripathi, Partha P. Talukdar: Improving Multi-hop Question Answering over Knowledge Graphs using Knowledge Base Embeddings. ACL 2020.
>
> [3] Gaole He, Yunshi Lan, Jing Jiang, Wayne Xin Zhao, Ji-Rong Wen: Improving Multi-hop Knowledge Base Question Answering by Learning Intermediate Supervision Signals. WSDM 2021.
>
> [4] Jiaxin Shi, Shulin Cao, Lei Hou, Juanzi Li, Hanwang Zhang: TransferNet: An Effective and Transparent Framework for Multi-hop Question Answering over Relation Graph. EMNLP 2021.
>
> [5] Belinda Z. Li, Sewon Min, Srinivasan Iyer, Yashar Mehdad, Wen-tau Yih: Efficient One-Pass End-to-End Entity Linking for Questions. EMNLP 2020.

---

### Meta-Review · Area_Chair_Txde · 2023-09-19

**Recommendation:** 4

**Metareview:**

This paper presents a mechanism for iterative reading then reasoning on large language models over structured data sources. A number of data source formats are considered, including SQL, tables and KG data. The method exploits the table / data structure as an interface  or type hint for information extraction and uses this information to return information from a linearized version of the data.

There is reasonable consistency with reviews, stating that the methodology is sound and has reasonable excitement. The method is dependent on the underlying LM. However, the methods in the paper describe an approach that should be agnostic to the underlying LM provider. It would be advantageous to also provide further experiments with open source LLMs as part of the final version. The disadvantage of only relying on OpenAI's implementation is (as demonstrated by the author's rebuttal), the final answer accuracy can be heavily dependent on the version of the model.

The reviewers identified several reasons to reject and spotted some useful areas that were missing in the paper, including the representation of the task as generating an intermediate query language. I would further suggest linking this to work in multi-step QA: for example the BREAK task: https://arxiv.org/abs/2001.11770 - regarding the point made by the reviewer about open source toolkits such as langchain, i believe that a module released by the authors of by this paper would be of significant value for the community and i would encourage the authors to release a their code.

---

### Decision · Program_Chairs · 2023-10-07

**Decision:**

Accept-Main

**Comment:**

This paper presents a mechanism for iterative reading then reasoning on large language models over structured data sources. A number of data source formats are considered, including SQL, tables and KG data. The method exploits the table / data structure as an interface  or type hint for information extraction and uses this information to return information from a linearized version of the data.

There is reasonable consistency with reviews, stating that the methodology is sound and has reasonable excitement. The method is dependent on the underlying LM. However, the methods in the paper describe an approach that should be agnostic to the underlying LM provider. It would be advantageous to also provide further experiments with open source LLMs as part of the final version. The disadvantage of only relying on OpenAI's implementation is (as demonstrated by the author's rebuttal), the final answer accuracy can be heavily dependent on the version of the model.

The reviewers identified several reasons to reject and spotted some useful areas that were missing in the paper, including the representation of the task as generating an intermediate query language. I would further suggest linking this to work in multi-step QA: for example the BREAK task: https://arxiv.org/abs/2001.11770 - regarding the point made by the reviewer about open source toolkits such as langchain, i believe that a module released by the authors of by this paper would be of significant value for the community and i would encourage the authors to release a their code.